# An Anti-Noise Convolutional Neural Network for Bearing Fault Diagnosis Based on Multi-Channel Data

**DOI:** 10.3390/s23156654

**Published:** 2023-07-25

**Authors:** Wei-Tao Zhang, Lu Liu, Dan Cui, Yu-Ying Ma, Ju Huang

**Affiliations:** 1School of Electronic Engineering, Xidian University, Xi’an 710071, China; 21021210622@stu.xidian.edu.cn (L.L.); 19021211299@stu.xidian.edu.cn (D.C.); 21021211282@stu.xidian.edu.cn (Y.-Y.M.); 2Research Institute of Guiyang Aero Engine Design Corporation of China, Guiyang 550081, China; 19021210807@stu.xidian.edu.cn

**Keywords:** fault diagnosis, convolutional neural network, three-dimensional (3-D) filter, bearing

## Abstract

In real world industrial applications, the working environment of a bearing varies with time, and some unexpected vibration noises from other equipment are inevitable. In order to improve the anti-noise performance of neural networks, a new prediction model and a multi-channel sample generation method are proposed to address the above problem. First, we proposed a multi-channel sample representation method based on the envelope time–frequency spectrum of a different channel and subsequent three-dimensional filtering to extract the fault features of samples. Second, we proposed a multi-channel data fusion neural network (MCFNN) for bearing fault discrimination, where the dropout technique is used in the training process based on a dataset with a wide rotation speed and various loads. In a noise-free environment, our experimental results demonstrated that the proposed method can reach a higher fault classification of 99.00%. In a noisy environment, the experimental results show that for the signal-to-noise ratio (SNR) of 0 dB, the fault classification averaged 11.80% higher than other methods and 32.89% higher under a SNR of −4 dB.

## 1. Introduction

As the core component of a rotating mechanism, the working state and health condition of a rolling bearing have a significant impact on the precision and stability of the equipment [1,2,3]. The spindle bearings of an aero-engine generally work under a high-intensity environment [4,5,6,7]. As part of a transmission joint, the spindle bearings are easily damaged. Once the rolling bearing fails, the performance of the engine will decline, which could lead to serious accidents. Therefore, it is necessary to predict the health of bearings, which can reduce maintenance costs, optimize resource allocation and ensure the normal operation of the equipment involved.

With the maturation of deep learning in recent years, various fields, such as image recognition, have also received opportunities to develop rapidly [8,9,10,11,12,13]. Deep learning can be applied in big data as a useful tool [14,15]. However, in deep learning, the feature extraction and fault classifier methods are often designed separately for traditional intelligent methods of fault diagnosis. Compared to traditional classification methods or classical equations [16,17,18,19], deep learning models can adaptively extract valuable features instead of relying on domain-specific knowledge accumulation, thereby improving the accuracy of diagnosis [20].

Currently, many neural network models have been proven to be applicable for fault-type diagnosis of bearings, such as RNN, CNN, and the typical BP network. During the past decade, a considerable number of works have been published on these methods. Shao et al. proposed the use of a deep auto-encoder model to extract features of different fault types from collected vibration signals [21]. They tried to feed samples with different sizes into the network and evaluated the maximum correntropy and artificial fish swarm on the performance. In some studies, the frequency domain features of signals are used as input samples for neural networks instead of using the raw features of the samples for the AE model. Jia et al. fed the frequency spectra of vibration signal instead of raw signals into SAE for rotating machinery diagnosis [22]. For example, as for rotating machinery diagnosis, Jia et al. tried to input the spectrum of vibration signals as a replacement for the raw signal for the network model, and the convergence speed and final accuracy of DNN models have been proven to be better by experimental results, compared with traditional neural network models (BPNN). By using a stacking restricted Boltzmann machine, DBN has also proven useful. Chen et al. established a domain adaptive DBN model with five hidden layers to obtain stronger robustness, which can achieve more stable performance under different working conditions, and the experimental results demonstrated that the average accuracy of classification is highest compared with other models under variable working conditions [23]. Tang et al. developed a bi-directional DBN to learn the fault features from the original vibration signals, which limits the impact of the dataset quality for the final accuracy [24]. Similar to optical image, convolutional neural network models are also good at extracting features from time–frequency samples, which are discussed in several studies, e.g., [25,26,27,28,29]. Specifically, the 1D CNN model has a strong advantage in analyzing time series data. For example, Zhang et al. proposed using a one-dimensional convolution network to diagnose the types of bearing faults, which performs well in resisting noise [30]. Liu et al. [31] proposed a novel method combining a 1D denoising convolutional auto-encoder (DCAE) and 1D CNN to promote the performance of the fault diagnosis method under a noisy environment. In addition, RNN network models are able to fully mine the information of temporal dimension of time series data which have unparalleled advantages in end-to-end fault diagnosis and are widely used in speech recognition. Zhang et al. [32] proposed a novel method based on RNN to identify fault types in rotating machinery that achieved the best performance and exhibited the robustness against the noise.

For fault diagnosis, it is necessary to use a small amount of data to learn more effective features so DL techniques can obtain better performance than traditional fault diagnosis methods [33,34,35]. With the improvement of technology [36], obtaining a large amount of data has become easier. Both traditional algorithms and deep learning algorithms can achieve good results. However, they still encountered the following drawbacks:The fault diagnosis models that are trained under a certain rotational speed usually fail when they are applied to the fault diagnosis of the same bearing under other rotational speeds;Existing models are often trained with samples formed by single-channel data. The samples formed using single-channel data contain fewer comprehensive features, which makes it difficult to result in a robust model for fault diagnosis;The training samples are usually noise-free samples that are obtained under better experimental conditions, so the resulting model will be bound to exhibit poor classification performance if practical noisy vibration data are directly tested without any denoising mechanism.

In our experiments, it is shown that even the state-of-the-art CNN fails to diagnose properly in noisy environment. Therefore, the existing models have not solved this problem yet.

In this paper, a novel fault diagnosis method based on multi-channel data fusion and MCFNN is proposed to address the problems above which is proven to have good performance. Our main contributions are as follows.

(1) To extract useful features for fault discrimination, we proposed a novel fault sample generation method based on the multi-channel envelope time–frequency spectrum of the original vibration signal and a subsequent three-dimensional (3-D) filtering. On one hand, we exploit the envelope time–frequency spectrum instead of short time Fourier transform of the original signal because it contains more obvious fault characteristic frequency components. On the other hand, 3D filtering manipulation smoothed the generated data samples from the time, frequency and channel dimensions, which greatly suppresses the background noise and other vibration pulse interference.

(2) We propose a novel CNN model, named MCFNN, which is used in fault diagnosis. In the part of feature extraction, there are three parallel convolutional layers with the kernels of different sizes for multi-scale feature extraction, two of which use the dropout mechanism with a continuously changing rate to enhance the anti-noise performance. Therefore, the model not only performs well under the noiseless environments, but also outperforms the state-of-art methods in noisy environments. Meanwhile, the result of feature visualization shows that the multi-scale feature extraction structure of MCFNN can obtain better feature clustering than DCNN and FDGRU.

This paper is organized as follows. In Section 2, the methodology of multi-channel sample generation is introduced. The intelligent diagnosis method based on MCFNN is described in Section 3. In Section 4, the experimental results are analyzed and discussed. Some experiments are conducted to evaluate the proposed method against other method. After this, a visualization of the diagnosis model is presented. Finally, general conclusions are given in Section 5.

## 2. Sample Generation for Model Learning

### 2.1. Data Normalization

When the bearing is damaged, the contact between the damaged surface and the engaging surface will produce a shock, which is usually represented as a transient pulse in vibration signal. The amplitude of the transient pulse is proportional to the rotational speed of the bearing, leading to significant variations in the amplitude of the vibration signal within the collected data. In the field of machine learning, the datasets with different magnitude are not conducive to data analysis.

To eliminate the effect of magnitude, data standardization is an effective way to eliminate differences between datas. After the normalization of the original data, the problems of prolonged training time and possible failure of network model convergence caused by singular data will be eliminated. We adopt a z-score standardization method, and the transformation function is
(1)$$s\left(t\right)=\frac{f(t)-\mu }{\sigma }$$
where f(t) represents the original vibration signal. μ is the mean of f(t). σ is the standard deviation of f(t), and s(t) is the vibration signal after normalization.

### 2.2. Envelope Time–Frequency Spectrum

The time-domain and frequency-domain of signals can be analyzed simultaneously by a time–frequency method such as short-time Fourier transform (STFT). It is worth noting that the envelope time–frequency spectrum differs from the conventional time–frequency transform is that it applies the FFT to the envelope of each frame signal, rather than directly on the frame signal itself. Compared with traditional time–frequency spectrum, the envelope time–frequency spectrum can suppress the interference signal and emphasize fault characteristics. The envelope signal of vibration signal can be written as
(2)$$e\left(t\right)=\sqrt{s_{R}^{2}\left(t\right)+s_{I}^{2}\left(t\right)}$$
where $s_{R}\left(t\right)=s\left(t\right)$ and $s_{I}\left(t\right)$ is the Hilbert transformation of s(t)
(3)$$s_{I}\left(t\right)=\frac{1}{\pi }\int_{-\infty }^{\infty }\frac{s_{I}\left(\tau \right)}{t-\tau }d\tau$$

As for time-varying non-stationary signals, STFT has certain advantages because of joint time–frequency analysis. It not only fully reflects the frequency information, but also effectively mirrors the time information by moving a time window. The basic operation formula is defined as
(4)$$STFT_{e}\left(\tau ,f\right)=\int_{-\infty }^{\infty }e\left(t\right)g\left(t-\tau \right)e^{-j2\pi ft}dt$$

An original signal is mapped two-dimensionally via inner product operation where e(t) is the envelope signal and $g\left(t-\pi \right)e^{-j2\pi ft}$ is the basis function of the STFT. The parameter *f* is the frequency of the Fourier transform.

According to the processing mechanism of STFT, the length of the window will determine the time and frequency resolution in the Fourier transformation. Specifically, the frequency resolution is directly proportional to the window size and inversely proportional to the time resolution. Therefore, choosing a suitable window length to balance the time resolution and frequency resolution is very important. The time resolution *T* and the frequency resolution *F* are calculated as
(5)$$T=\left[\frac{N_{x}-N_{o}}{N_{w}-N_{o}}\right]$$
(6)$$F=\left[\frac{N_{f}}{2}+1\right]$$
where $\left[\cdot \right]$ means rounding down. $N_{x}$ represents the length of a single sample. $N_{w}$ is the width of the time window. $N_{o}$ is the number of overlapping points, and $N_{f}$ is the number of points participating in the Fourier transform. The resolution *P* of the time–frequency spectrum is defined as
(7)P=T×F

Figure 1 illustrates the calculation process of obtaining the input samples for neural networks and displays the resulting images at each stage. First, the signal used for STFT and normalization can be obtained by segmenting the original signal whose duration is 0.1 s as an experience value. Second, we standardize each signal sample by using the z-score normalization. Finally, the processed signal is transformed into the time–frequency spectrum of the envelope signal, thus obtaining the feature representation of the sample sequence.

### 2.3. 3-D Filter

The environment of the bearing is noisy, and the angle-contacting bearings change with the bearing speed, which leads to the drift of the fault characteristic frequency. The envelope time–frequency spectrum distribution of the vibration signal is different even for the same fault type. Therefore, the envelope time–frequency spectrum of the signal cannot represent features correctly in some conditions. Further processing of the envelope time–frequency spectrum is needed. The three-dimensional (3D) filter is introduced to process the envelope time–frequency spectrum, which is the three-dimensional samples. This approach effectively eradicates noise, enhances the concentration of spectral lines, and fortifies the interconnection between data from each channel. In addition, the three-dimensional data of each channel contains fault characteristic information from other channels, which makes the fault information of multi-channel fault samples more abundant.

The filter is a linear smooth filter whose weights satisfy Gauss distribution. It has a good suppression effect on the noise following normal distribution. The expression of the three-dimensional Gaussian function is
(8)$$g\left(x,y,z\right)=\exp\left(-\frac{x^{2}+y^{2}+z^{2}}{2\sigma^{2}}\right)$$
where g(x,y,z) represents the weight of the three-dimensional Gaussian filter (x,y,z). *x*, *y*, *z* represent values of different dimensions, respectively, and σ is the variance of the Gaussian function.

The three-dimensional Gaussian function is non-negative in the whole domain, so it is an infinite convolution kernel. However, such a large Gaussian kernel is not required in actual situations. Generally, we only need to set the value within 3σ and then select a Gaussian kernel with an appropriate size based on actual application scenarios.

In this paper, a 3 × 3 × 3 Gaussian filter is used to filter the fusion data of the three channels. Similarly, a 8 × 8 × 8 Gaussian filter is used to filter the fusion data of the eight channels. The first part of Figure 2 is a schematic diagram of a 3D filtering process for 3D inner fault samples formed by channels 0, 1 and 7.

## 3. Model Design

### 3.1. The Proposed Network

After collecting the bearing signals emanating from eight sensors strategically placed at different locations amid varying rotational speeds, a series of technologies, such as truncation, normalization, Short-Time Fourier Transform (STFT) transformation, and Gaussian filtering, are used on the original signal sequentially. The sample for the network can be obtained by the above processes. Then, the MCFNN model is built and trained using a small portion of multi-channel samples corresponding to a certain rotational speed. At last, the trained model is applied to the rest of multi-channel samples that corresponds to other rotational speed to test model performance, and the visualization of the learning process of MCFNN model via the t-SNE technique shows some insights into the model behavior.

The proposed MCFNN model consists of six convolutional layers, two addition layers, one Concat layer, two pooling layers, and a Softmax layer. The pooling layer is max polling and the activation function is ReLU. The batch normalization is designed after each convolutional layer to prevent the gradient explosion and speed up the convergence of MCFNN. It is noteworthy that there are three parallel convolutional layers with the kernels of different sizes for multi-scale feature extraction, two of which use the dropout mechanism with continuously changing rate to improve the anti-noise performance of neural networks. The parameters of each layer are detailed in Table 1, where *c* represents the number of channels. The experiments are implemented under the PyTorch platform.

### 3.2. Learning Procedure of MCFNN

In the proposed model, the CNN network is mainly responsible for extracting features from samples, while the fully connected layer is responsible for classifying to obtain the final fault type. The learning procedure of MCFNN is mainly summarized as two stages: forward calculation and parameter reverse update.

(1)Convolutional layer

The convolutional layer greatly reduces the number of parameters by introducing the local connection and weight sharing. The calculation method of convolution is as follows:(9)$$a_{rst}^{l}=f\left(\sum_{i=0}^{W-1}\sum_{j=0}^{W-1}\sum_{k=0}^{K_{l-1}-1}w_{ijkt}^{l}a_{\left(r-i)(s-j)(k\right)}^{l-1}+b_{t}^{l}\right)$$
where $r=1,\ldots ,R_{l}$, $s=1,\ldots ,S_{l}$, $t=1,\ldots ,K_{l}$, l=1,…,L, $R_{l}$, $S_{l}$, and $K_{l}$ represent the size of the *l*th feature map respectively. $a_{rst}^{l}$ indicates the (r,s,t)th output on the *l*th convolutional layer. $w_{ijkt}^{l}$ represents the *t*th convolution kernel of *l*th convolutional layer. $b_{t}^{l}$ denotes as the bias of the *t*th filter of *l*th convolutional layer. *W* is the width of the kernel, $K_{l-1}$ is the number of feature map on l−1th convolutional layer. The activation function f(x) should be used after each convolution operation to enhance the nonlinearity of the network.
(10)$$f\left(x\right)=\left\{\begin{array}{c} x,\;x>0\\ 0,\;x\leq 0 \end{array}\right.$$

Dropout is used in the convolution kernel of multi-branch feature extraction, which adds certain interference during training model. The parallel convolution kernels can extract better feature maps of different scales, which is beneficial for the network to learn more feature information and complete fault diagnosis more accurately.

To alleviate the overfitting phenomenon of the network, Srivastava et al. [37] proposed the dropout method. The core idea is to randomly deactivate neurons with a certain probability *p* in the training process, which prevents the output results from overly relying on a few specific features. Compared with regularization methods, the experimental results show that dropout method has more advantages in enhancing the generalization of the network.

Srivastava et al. mentioned that some parameters can be lost randomly in all multilayer perceptron layers to obtain better performance [37]. Because of the fewer parameters in the convolutional kernel, the loss of parameters randomly has a significant impact on the final results. So, we propose adding dropout into the convolution kernel of multi-branch feature extraction, thus adding noise to the convoluted area. Our third-layer and fourth-layer kernels are different, which makes dropout more effective as different levels of noise are added at the same time. If the same kernel size is used, there will only add the same extent of noise simultaneously under a fixed dropout rate. In real working conditions, the signal may be mixed with varying degrees of noise. To fit the actual situation, different dropout rates are used for each batch training which are random values between 0.1∼0.9.
(11)p∼Uniform(0.1,0.9)

$\xi_{ijkt}^{l}$ follows Bernoulli distribution, which is used to decide whether the (i,j,k)th weight in the *t*th convolution kernel of the *l*th layer is dropped or not. According to the above description, when dropout is used on the convolution kernel of a certain layer, the convolution process of this layer can be expressed as
(12)$$\xi_{ijkt}^{l}∼Bernoulli\left(p\right)$$(13)$${\tilde{w}}_{ijkt}^{l}=\xi_{ijkt}^{l}w_{ijkt}^{l}$$(14)$$a_{rst}^{l}=f\left(\sum_{i=0}^{W-1}\sum_{j=0}^{W-1}\sum_{k=0}^{K_{l-1}-1}w_{ijkt}^{l}a_{\left(r-i)(s-j)(k\right)}^{l-1}+b_{t}^{l}\right)$$
where ${\tilde{w}}_{ijkt}^{l}$ represents the (i,j,k)th weight after the dropout operation in the *t*th convolution kernel of the layer *l*.

(2)Batch Normalization

Batch Normalization (BN), used after the Rectified Linear Unit (ReLU) and incorporated within each convolutional layer, facilitates the expeditious convergence of the neural network model and mitigates the risk for gradient explosion. The BN layer can be expounded upon as follows:(15)$${\hat{a}}_{rst}^{l}=\frac{a_{rst}^{l}-\mu_{B}}{\sqrt{\left(\sigma_{B}^{2}+\varepsilon \right)}}$$
(16)$$y_{rst}^{l}=\gamma_{t}^{l}{\hat{a}}_{rst}^{l}+\beta_{t}^{l}$$
where $\mu_{B}=E\left[a_{rst}^{l}\right]$, $\sigma_{B}^{2}=Var\left[a_{rst}^{l}\right]$, and $y_{rst}^{l}$ is the (r,s,t)th output on the *l*th BN layer. σ is a small constant that increases numerical stability. $\gamma_{t}^{l}$ and $\beta_{t}^{l}$ are the scale and shift parameters to be learned, respectively.

(3)Pooling layer

The maximum pooling layer can effectively select the most valuable features and reduce the complexity of the model, which is written as:(17)$$p_{rst}^{l}=\max_{(r-1)Q+1\leq x\leq rQ,(s-1)Q+1\leq y\leq sQ}\left\{a_{xyt}^{l}\right\}$$
where $p_{rst}^{l}$ expresses the corresponding value of the neuron in layer *l* of the pooling arithmetic. *Q* is the window size of pooling layer. $a_{xyt}^{l}$ denotes the element of (x,y)th neuron in the *t*th page of layer *l*.

(4)Fully-connected layer

After several rounds of feature extraction layers, the feature will be through a fully connected layer, which is also known as the classification layer. The Softmax regression is commonly selected as the last layer to complete the multi-classification problems. Then, the softmax function is shown as follows:(18)$$q_{j}=soft\max\left(z_{j}\right)=\frac{e^{z_{j}}}{\sum_{c=1}^{C}e^{z_{c}}},\;j=0,\ldots ,C-1$$
where $z_{j}$ express the value of the *j*th output neuron, *C* is the number of fault types.

The cross-entropy loss function can intuitively reflect the difference between the model output results and the labels, thereby mirroring the performance of the model. The cross-entropy function is defined as follows:(19)$$E=-\frac{1}{M}\sum_{m=1}^{M}\sum_{j=0}^{C-1}1\left\{y_{m}=j\right\}\log q_{j}$$
where *M* is the total number of samples, 1{·} represents the indicator function which can only equal 1 or 0, where the condition in parentheses returns 1. $y_{m}$ is the true label of *m*th sample, $y_{m}\in \left\{0,\ldots ,C-1\right\}$.

During the training process, we need to update the parameters of convolutional layers and BN layers by back-propagation. The chain rule for BN layers is described as follows:(20)$$\frac{\partial E}{\partial {\hat{a}}_{rst}^{l}}=\frac{\partial E}{\partial y_{rst}^{l}}\gamma_{t}^{l}$$
(21)$$\frac{\partial E}{\partial \sigma_{B}^{2}}=-\frac{1}{2}\sum_{k=1}^{n}\sum_{r=0}^{R_{l}-1}\sum_{s=0}^{S_{l}-1}\sum_{t=0}^{K_{l}-1}\frac{\partial E}{\partial {\hat{a}}_{rst}^{l}\left(k\right)}\left(a_{rst}^{l}\left(k\right)-\mu_{B}\right){\left(\sigma_{B}^{2}+\varepsilon \right)}^{-3/2}$$
(22)$$\frac{\partial E}{\partial \mu_{B}}=-\sum_{k=1}^{n}\sum_{r=0}^{R_{l}-1}\sum_{s=0}^{S_{l}-1}\sum_{t=0}^{K_{l}-1}\frac{\partial E}{\partial {\hat{a}}_{rst}^{l}\left(k\right)}\frac{1}{\sqrt{\left(\sigma_{B}^{2}+\varepsilon \right)}}-2\frac{\partial E}{\partial \sigma_{B}^{2}}\sum_{k=1}^{n}\sum_{r=0}^{R_{l}-1}\sum_{s=0}^{S_{l}-1}\sum_{t=0}^{K_{l}-1}\frac{\partial E}{\partial {\hat{a}}_{rst}^{l}\left(k\right)}\frac{\left(a_{rst}^{l}\left(k\right)-\mu_{B}\right)}{nR_{l}S_{l}K_{l}}$$
(23)$$\frac{\partial E}{\partial a_{rst}^{l}\left(k\right)}=\frac{\partial E}{\partial {\hat{a}}_{rst}^{l}\left(k\right)}\frac{1}{\sqrt{\left(\sigma_{B}^{2}+\varepsilon \right)}}+2\frac{\partial E}{\partial \sigma_{B}^{2}}\frac{\left(a_{rst}^{l}\left(k\right)-\mu_{B}\right)}{nR_{l}S_{l}K_{l}}+\frac{\partial E}{nR_{l}S_{l}K_{l}\partial \mu_{B}}$$
(24)$$\frac{\partial E}{\partial \gamma_{t}^{l}}=\sum_{k=1}^{n}\sum_{r=0}^{R_{l}-1}\sum_{s=0}^{S_{l}-1}\frac{\partial E}{\partial y_{rst}^{l}\left(k\right)}{\hat{a}}_{rst}^{l}$$
(25)$$\frac{\partial E}{\partial \beta_{t}^{l}}=\sum_{k=1}^{n}\sum_{r=0}^{R_{l}-1}\sum_{s=0}^{S_{l}-1}\frac{\partial E}{\partial y_{rst}^{l}\left(k\right)}$$
where *k* denotes the *k*th sample in the training mini-batch, and *n* is the size of the mini-batch.

When $\frac{\partial E}{\partial a_{\mathrm{rst}}^{l}}$ is computed, the gradient of convolutional kernels can be calculated by the chain rule of differentiation, which is described as follows:(26)$$\frac{\partial E}{\partial \omega_{xyzt}^{l}}=\sum_{k=1}^{n}\sum_{r=0}^{R_{l}-1}\sum_{s=0}^{S_{l}-1}\frac{\partial E}{\partial a_{rst}^{l}\left(k\right)}a_{\left(r-x\right)\left(s-y\right)z}^{l-1}$$
(27)$$\frac{\partial E}{\partial b_{t}^{l}}=\sum_{k=1}^{n}\sum_{r=0}^{R_{l}-1}\sum_{s=0}^{S_{l}-1}\frac{\partial E}{\partial a_{rst}^{l}\left(k\right)}$$ The parameters are updated using gradient descent learning as follows:(28)$$w_{ijkt}^{l}\left(h\right)=w_{ijkt}^{l}\left(h-1\right)-\eta \frac{\partial E}{\partial w_{ijkt}^{l}}$$
(29)$$b_{t}^{l}\left(h\right)=b_{t}^{l}\left(h-1\right)-\eta \frac{\partial E}{\partial b_{t}^{l}}$$
where η is the learning rate, and *h* is the iteration index.

## 4. Experiment Validation

In the actual production process, the generation of noise is influenced by multiple factors. Even if a large number of noise samples are collected under different conditions, it is difficult to ensure that the samples in this dataset can represent all application scenarios in real life. However, it is feasible to collect vibration signals without noise under specific experimental conditions. So, it is necessary to train a network with strong robustness by using a noise-free dataset.

### 4.1. Data Acquisition

The bearing data used in this study was collected on SB25 aeroengine bearing testbed through eight vibration sensors, whose numbers were AC0 to AC7 in turn, as shown in Figure 3. In Figure 3a, AC0, AC1 and AC7 are placed on a housing of bearing pedestal. In Figure 3b, the other five sensors are placed on the outer shell of the testbed. The parameters of the bearing were shown in Table 2. Since the contact angle changes with speed, the actual contact angle in Table 2 is a dynamic range. The dataset is collected from the machine under six different loads (4, 5, 6, 7, 8, and 9 kN). The speed range of the bearing is 1000 rpm to 10,000 rpm, the sampling frequency is 20 kHz, and the data duration is 10 s. The bearing dataset consisted of the following five conditions: (1) normal condition (NC), (2) with inner race fault (IF), (3) with outer race fault (OF), (4) with ball fault (BF), and with cage fault (CF). The type of fault is presented in Figure 4.

In this study, there are 12 training datasets according to the different sensor positions and combinations between different channels, namely Sr0, Sr1, Sr2, Sr3, Sr4, Sr5, Sr6, Sr7, Mr0, Mr1, Mr2 and Fr0 datasets, which are used to train the MCFNN model, respectively. As a result, various MCFNN models corresponding to single-channel, multi-channel and full-channel data are obtained. The details of the datasets are displayed in Table 3. The single-channel datasets Sr0, Sr1, Sr2, Sr3, Sr4, Sr5, Sr6 and Sr7 are collected from sensors AC0 to AC7, respectively, each of which includes five fault categories under loads of 4, 5, 6, 7, 8, and 9 kN. Each category contains 27,300 samples, and the size of each sample is 64 × 64 × 1. Multi-channel datasets Mr0, Mr1 and Mr2 are three different fusions of the above single-channel datasets via the concatenation from the third mode. For example, the fusion of Sr0, Sr1, and Sr7 corresponding samples along the third dimension yields dataset Mr0. Therefore, the size of each sample of datasets Mr0, Mr1 and Mr2 is 64 × 64 × 3. Dataset Fr0 is a deep fusion of datasets Sr0 to Sr7 so that the size of each sample is 64 × 64 × 8.

As for all training datasets, the samples are collected under 46 different rotational speeds, such as 1000 rpm, 1200 rpm and 10,000 rpm, ranging from 1000 rpm to 10,000 rpm with an equal interval step of 200 rpm. Different from training datasets, the speed range of the test datasets is from 1100 rpm to 9900 rpm, which include 45 different rotational speeds. The training set and test set are non-overlapping with each other in rotational speed. Finally, we collect 69,000 samples for each training dataset and 67,500 samples for each test dataset.

### 4.2. Input Image Size

In CNN, the size of the input sample has an important impact on the training time and forward prediction time of the network. At the same time, the fault information contained in the time–frequency spectrum with different resolutions is different. If the resolution is low, the time–frequency spectrum energy distribution of different fault types will be similar, which is not conducive for the network to complete the classification. However, when the resolution is high, the prediction time of the network will increase, which cannot meet the real-time requirements. Therefore, a balance needs to be struck between real-time and effectiveness. Different from the Fourier points, overlapping ratio and window length are used to obtain samples with different time–frequency. When the time–frequency resolution is odd, the last row and column of the sample were abandoned to facilitate the operation of the pooling layer. Dataset Mr0 and Me0 are chosen to test the proposed method to obtain a better sample input size. The input sample sizes of CNN are listed in Table 4.

The diagnosis results are shown in Figure 5. In this figure, the accuracy is calculated in the test set, and the time indicator includes the training time and testing time. The test accuracy is greater than 97%, which shows that feature extraction based on multi-channel samples. When the input sample size is 16 × 16 × 3, the fault diagnosis accuracy is 97.6%. When the input sample size is 32 × 32 × 3, the fault diagnosis accuracy is improved by 1.9% compared to when the input sample size is 16 × 16 × 3, and the training time of the network increases by 8.5 s. When the input sample size is 64 × 64 × 3 and the window length is 128, the fault diagnosis accuracy is 100%, which is 0.2% higher than that of the window length, which is 64, and the training time is the same for both. Compared with the input sample size of 32 × 32 × 3, the training time of the network increases by 98 s on average. When the input sample size is 128 × 128 × 3, the diagnosis accuracy is up to 100%, but the training time of the network is 3.2 times that of the input sample size is 64 × 64 × 3. We observe a steep increase in time with a decrease in the size of the sample. When the window length of 64 and 128, the accuracy of the model has almost reached 100%, and it is meaningless to continue increasing the size of the sample. Therefore, in the following experiments, we will choose the window length of 64. Figure 6 shows the envelope time–frequency images of a bearing operating in five different states.

### 4.3. Selection of Learning Rate

In the training process of CNN, gradient descent method is often used to update the network parameters, in which the learning rate is an important parameter affecting the convergence result. Choosing the appropriate learning rate is of great use to accelerate the convergence speed of network training and guarantee the stability of output results. In this paper, the fault diagnosis accuracy and loss function value in the process of network training and testing are calculated at learning rates of 0.0001 to 0.03 (other parameters in the network remain unchanged, and batch size is taken as 100). During the training process, the programming language is Python. The data processing computer is equipped with Intel E5-2620 × 2 CPU, GTX Titan Xp GPU and 128 GB memory. The specific results are shown in Table 5.

It can be seen from Table 5 that with the increase in the learning rate, the fault diagnosis accuracy of bearing fault shows a downward trend. Considering the loss function value, fault diagnosis accuracy and network training time, the learning rate is taken as 0.001 in this paper.

### 4.4. Diagnosis Results under Noise-Free Environment

The twelve training set described in Table 3 are used to train the fault diagnosis network, whose structure is shown in Table 1, and the effectiveness of the models are evaluated with the test sets. Since the neural network is randomly initialized, in order to ensure the reliability of the model, 20 trails are carried out for each bearing dataset. The diagnosis results are shown in Figure 7. It can be seen from Figure 7i,l that the diagnostic accuracies are 100% for datasets Me0 and Fe0. For dataset Me1, diagnostic accuracies of 100% are obtained five times, and accuracies are greater than 99.6% in fifteen cases. Diagnostic accuracies of 100% are obtained six times by dataset Me2, accuracies of 99.9% are obtained eight times, and accuracies of 99.8% are obtained six times. The experimental results indicate that the performance of the proposed network model is highly reliable.

For comparison, we select the single-channel envelope time–frequency spectrum as the input of the proposed model. For the proposed model, it can be seen from Figure 7a that the diagnostic accuracies greater than 98% are obtained 16 times for dataset Se0. Dataset Se1 obtains diagnostic accuracies greater than 97.3% in seventeen cases and greater than 96.1% in three cases. For dataset Se2, the highest diagnostic accuracy is 96.1%, and the lowest is 88.2%. Dataset Se3 obtains diagnostic accuracies greater than 96%. Diagnostic accuracies are greater than 96% in 16 cases by dataset Se4. After 20 times testing, the highest diagnostic accuracy of dataset Se5 is 97.5% and the lowest is 95.1%. The fault diagnosis accuracies of dataset Se6 fell in the range of 98–99% for 10 times, and the rest are higher than 96%. For dataset Se7, diagnostic accuracies greater than 98.1% are obtained sixteen times, and accuracies greater than 95.6% are obtained four times. It is obvious seen from Figure 7a–h that the diagnostic accuracy of single-channel training set is less than 100% with large fluctuation, illustrating that the fault diagnosis model using single-channel has poor stability. The comparison result of single-channel and multi-channel shows that the multi-channel bearing fault diagnosis method has a better diagnostic performance for the five fault types of bearings, and the model has strong stability and generalization ability.

To verify the superiority of the proposed method, the method is compared with the deep convolutional network (DCNN) based on continuous wavelet transform (CWT) [38] and the model based on Gated Recurrent Unit (GRU) for fault diagnosis (FD). The DCNN model contain three convolutional layers and one fully connected layer. The convolution kernel size of the three layers is as follows: 5 × 5 × 12, 5 × 5 × 12, and 7 × 7 × 12. There are three neurons in the fully connected layer, which are changed to five in this paper corresponding to the five kinds of health state. The input samples of DCNN are time–frequency spectrum after using CWT for vibration signals. For the twelve datasets described in Section 4.1, the CWT+DCNN algorithm is evaluated by 20 independent experiments. The diagnosis results using CWT+DCNN are shown in Figure 7. The diagnostic accuracy of datasets Se0 and Se1 is about 97%. The average diagnostic accuracies of 20 trails of datasets Se6 and Se7 are about 94%. The diagnostic accuracy of datasets Se2 to Se5 is relatively low, and the lowest is about 62%. Figure 7h,i shows that the accuracy of multi-channel fault diagnosis is almost 99%. As for FDGRU [32], the samples obtained by signal-to-image conversion for the original signal are inputted into the GRU network firstly and then the multilayer perceptron (MLP) is used to further realize fault identification. The size of consecutive signal segment is 64 and the hidden size of MLP is 1024. Figure 7 shows that the accuracy of FDGRU is similar to that of the CWT+DCNN model.

Table 6 shows the standard deviation and the average accuracy of 20 trails for each dataset, which further demonstrate the stability of the proposed method. All models trained by a multi-channel dataset perform better, compared the model using single-channel data. The decrease in the average test accuracy of single-channel diagnosis based on the comparison method is larger, especially in datasets Se2 to Se5, but the proposed method has an average test accuracy greater than 93%. There are two main reasons for this phenomenon. The first reason is that the sensors corresponding to channels 2 to 5 are all arranged on the shell of the testbed, which is far from the fault source. The frequency components of the collected vibration signals are complex due to the environmental effect. The generated two-dimensional image samples increase the difficulty of feature learning and are prone to misjudgment. At the same time, the structure of DCNN and FDGRU are relatively single, and the information of the feature map extracted by the network is seriously missing, which leads to the decline of fault diagnosis performance. In conclusion, the performance of the proposed method in single-channel and multi-channel fault diagnosis is better than the CWT+DCNN and FDGRU algorithms.

To demonstrate the superiority of multi-channel algorithms, the improvement rate of multi-channel fault diagnosis accuracy compared with single-channel fault diagnosis accuracy is calculated. The diagnosis accuracy of the dataset Me0 is higher than that of every single channel, and the standard deviation of the average fault diagnosis accuracy is 0, which is 1.24% lower than that of single-channel fault diagnosis models (Se0, Se1, Se7), indicating that the stability of a multi-channel fault diagnosis model is better. The diagnosis accuracy based on a channel 2,3,4 data fusion is 6.6% higher than that of a single channel, and the fault diagnosis accuracy of the dataset Me2 is 2.2% higher than that of a single channel. Beyond all doubt, the fault diagnosis method with multi-channels has a better performance.

Figure 8 shows the error curve in the training process when using the proposed method, CWT+DCNN and FDGRU trained with the twelve datasets described in Section 4.1, respectively. It can be seen from Figure 8 that the training error of the proposed method is almost close to 0 after 300 iteration training, and the initial error value is small. However, using the CWT+DCNN algorithm to train the training sets of channels 2 to 5, respectively, it is difficult to continue to decline when the error drops to a certain value. Although the error curve of multi-channel fault diagnosis using the CWT+DCNN algorithm converges faster than that of single-channel fault diagnosis, there is still a certain gap compared with the proposed method.

### 4.5. Performance under Noisy Environment

The anti-noise performance of the model is discussed in this section. In our experiments, the model is trained using the training set, and is tested on a noisy dataset. This condition is closer to the real-world condition. Therefore, it is a feasible scheme to use the original signal added with Gaussian white noise with different intensities as the noise sample with different SNR is defined as
(30)$$SNR_{dB}=10\log_{10}\left(\frac{P_{signal}}{P_{noise}}\right)$$
where $P_{signal}$ and $P_{noise}$ are the power of signal and the noise, respectively.

We test the proposed method using noisy signals with a range of −4 dB to 10 dB. Figure 9 shows the original signal and the noise signal with −4 dB.

The results of the three methods diagnosing with noisy signals are shown in Figure 10. Figure 10 compares the fault diagnosis accuracy of proposed method, CWT+DCNN and FDGRU algorithm trained with different input samples under different SNR. Figure 10 illustrates that when trained with 69,000 noise-free training samples, all algorithms achieve nearly 100% accuracy on test samples with high SNR. However, as the SNR decreases, the accuracy of the CWT+DCNN algorithm also suffers from a remarkable decrease. When the SNR of the datasets Se0 and Me0 equals 0 dB, the accuracies of the CWT+DCNN algorithm are 65.98% and 90.67%, respectively, the FDGRU algorithm are 62.17% and 81.99%, respectively, and the accuracies of the proposed method are 92.49% and 97.91%, respectively. The accuracy of the proposed network based on single-channel fault diagnosis is 1.82% higher than that of CWT+DCNN algorithm and 10.5% higher than that of FDGRU algorithm for multi-channel fault diagnosis. In multi-channel fault diagnosis, the diagnosis accuracy of the proposed method is 7.24% higher than that of CWT+DCNN algorithm and 15.92% higher than that of FDGRU algorithm. In single-channel fault diagnosis, the accuracy of the proposed method is 26.51% higher than that of CWT+DCNN algorithm and 30.32 higher than that of FDGRU algorithm. When SNR equals −4 dB, the accuracy of dataset Se0 is 76.12% for the proposed method, which is smaller than 87.67% for dataset Me0. As SNR decreases, the performance of the proposed method with single-channel data is better than that of the CWT+DCNN algorithm and FDGRU algorithm based on multi-channel fault diagnosis. The reason for this phenomenon can be summarized as two points: the structure of the fault diagnosis network and sample generation. From the perspective of the network structure, the structure of DCNN is relatively single, and the size of the convolution kernel increases layer by layer. The feature map extracted in the deep layer of the network omits a lot of detailed information. However, the MCFNN uses dropout to randomly zero the convolution kernel for multi-branch feature extraction, which enhances the anti-interference performance of the network. For FDGRU networks, it can fully utilize the temporal information of samples, but cannot make full use of the information between channels due to the limitations of GRU networks. From the perspective of sample generation, for the three-dimensional samples formed by CWT+DCNN algorithm, the samples between multiple channels exist independently without any associated information. In order to make multi-channel data able to be input into the FDGRU network, we use a convolutional layer with a 1 × 1 kernel to adjust the number of channels. However, the single-layer linear 1 × 1 convolutional layer only maps features from high-dimensional to low-dimensional, and cannot utilize information between channels. The 3D filter used in the proposed method strengthens the correlation between multiple channel data. After filtering, the samples of a single channel contain the fault information of other channel samples, which can suppress noise to a certain extent and enhance the fault feature expression ability of multi-channel samples. The experimental results indicate that the proposed method has higher advantages in terms of both noise resistance and classification accuracy.

Figure 11 shows the confusion matrix of the test results of the proposed method, CWT+DCNN and FDGRU algorithm on dataset Me0 with SNR of −4 dB and 0 dB. Note that, when SNR equals −4 dB of dataset Me0, the proposed method can diagnose other fault types except the inner fault with a high accuracy, and the diagnosis accuracy of inner fault is only 78.7%. By using CWT+DCNN algorithm, the accuracies of inner and cage fault are 34.7% and 58.8%, respectively. As for the FDGRU model, the accuracy of normal is only 12.6%. When SNR of dataset Me0 equals 0 dB, the proposed method can successfully diagnose five fault types and the accuracy exceeds 95%. The diagnosis accuracy of CWT+DCNN algorithm for inner fault has reached 84.5%, and the accuracy of the cage fault is 92.6%. Among these three models, the FDGRU model has the highest accuracy in the outer faults, but the overall accuracy is low. This suggests that the proposed method improves the diagnosis accuracy of five bearing fault types at low signal-to-noise environments.

### 4.6. Model Visualization

We visualize the feature maps of some layers of the network to determine whether the neural network has extracted recognizable features. T-SNE intuitively reflects the clustering of features by compressing high-dimensional data into two dimensions [39].

Visualization of the feature extracted from partial layers of the network using the dataset Me0 with SNR = 0 dB is shown in Figure 12. It is not difficult to find that the features of different types of faults become more divisible with the deepening of the network layers. This suggests that the constructed multi-branch structure can effectively extract the characteristics of samples under noisy environment. Secondly, as we can see from the visualization of fc layer, the feature points of inner and ball fault have little distance, which suggests that the model may not perform very well in discriminating between inner and ball fault. At the same time, the feature points of inner and outer fault have some overlapped region. The visualization results illustrate that the diagnosis accuracy of MCFNN decreases in strong noise, mainly due to the weak judgment ability of these three fault types. After observing the visualization results of FDGRU, it is not difficult to find that the overlapped region is large relatively, and that means that the clustering effect of the FDGRU model is relatively poor, compared with other methods. When the noise intensity increases, the performance of the model also rapidly decreases.

The right side of Figure 12 shows the feature visualization map of all models. The distance of the inner, ball and outer groups is small, which indicates that the DCNN model may not perform well in distinguishing these fault types. According the comparison results based on the feature visualization of the three methods, it can be shown that the proposed method can better distinguish different fault types under the noisy environment.

## 5. Conclusions

In this paper, a novel fault diagnosis method based on multi-channel data and MCFNN is proposed for bearing fault diagnosis. Firstly, the multi-channel envelope time–frequency spectrum is filtered by 3D filtering to generate the input samples with more obvious fault characteristics. Secondly, MCFNN is designed to complete the fault diagnosis, which includes three parallel convolutional layers with different sizes in the filter stages, two of which use dropout with a constantly changing rate to enhance the anti-noise ability of the model. With the help of the generated 3D fault samples, the proposed MCFNN model works well under a noisy environment and performs successfully when the working condition changes. In addition, network visualizations are used to explore the reasons for the high performance of proposed MCFNN model in feature extraction and classification.

The experimental results in the Section 4 indicate that the model has superior anti-noise performance with the multi-channel dataset, compared with popular CNN models. In the noisy environment, for the signal-to-noise ratio (SNR) of 0 dB, the fault classification is averaged 11.80% higher than other methods and 32.89% higher under SNR of −4 dB. In future work, the author plans to simplify and improve existing models while maintaining their performance to enable easy deployment on resource-constrained embedded devices, enhancing their practicality in engineering applications.

## Figures and Tables

**Figure 1 sensors-23-06654-f001:**
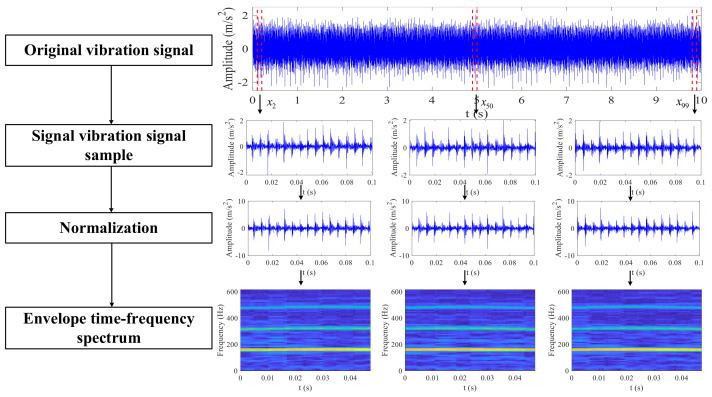
Two-dimensional image representation based on STFT.

**Figure 2 sensors-23-06654-f002:**
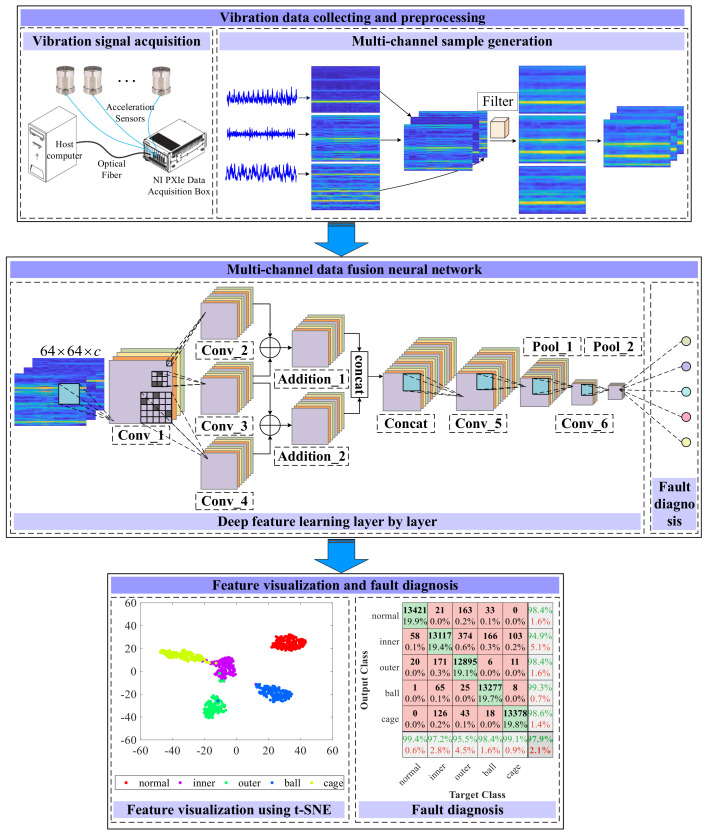
The overall fault diagnosis framework based on multi-channel samples and MCFNN.

**Figure 3 sensors-23-06654-f003:**
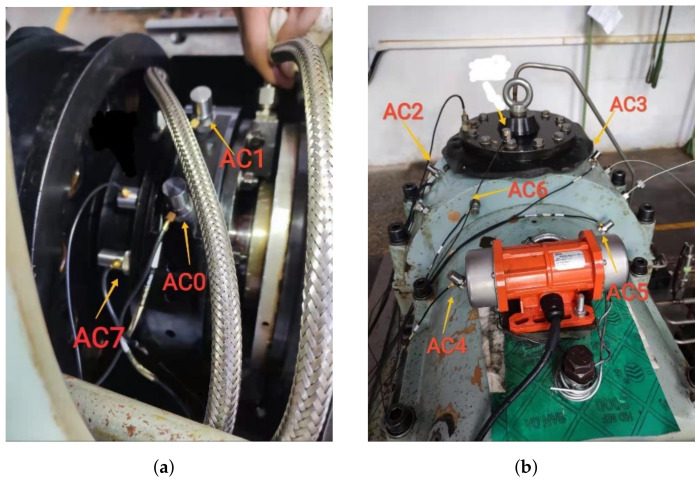
Sensor position distribution. (**a**) Housing of bearing pedestal. (**b**) Enclosure of testbed.

**Figure 4 sensors-23-06654-f004:**
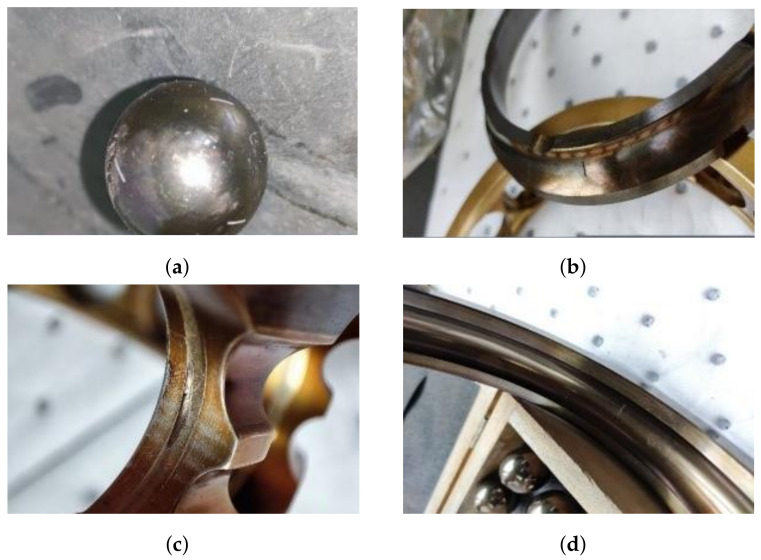
Four fault types. (**a**) Ball fault (BF). (**b**) Outer fault (OF). (**c**) cage fault (CF). (**d**) inner fault (IF).

**Figure 5 sensors-23-06654-f005:**
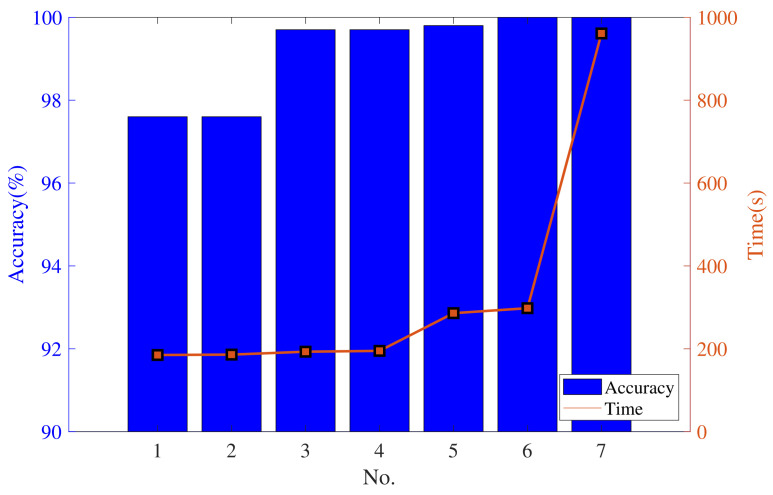
Average accuracy and time with different input sample sizes.

**Figure 6 sensors-23-06654-f006:**

Envelope time–frequency images of bearing under different health conditions: (**a**) normal condition, (**b**) inner race fault, (**c**) outer race fault, (**d**) ball fault, and (**e**) cage fault.

**Figure 7 sensors-23-06654-f007:**
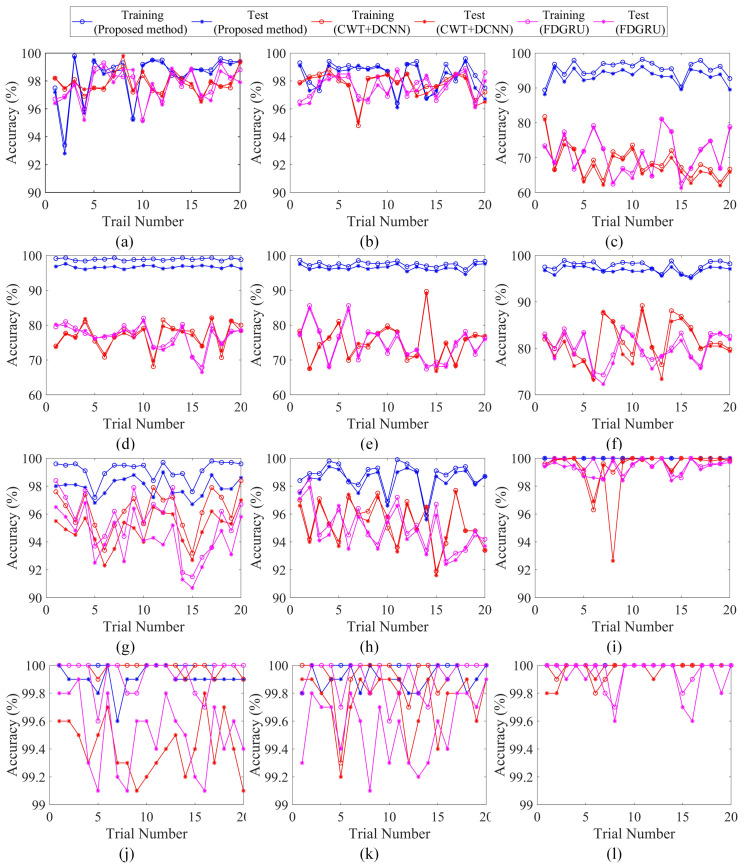
Diagnosis results of 20 trials of bearing datasets using three methods. (**a**) Dataset Sr0, Se0. (**b**) Dataset Sr1, Se1. (**c**) Dataset Sr2, Se2. (**d**) Dataset Sr3, Se3. (**e**) Dataset Sr4, Se4. (**f**) Dataset Sr5, Se5. (**g**) Dataset Sr6, Se6. (**h**) Dataset Sr7, Se7. (**i**) Dataset Mr0, Me0. (**j**) Dataset Mr1, Me1. (**k**) Dataset Mr2, Me2. (**l**) Dataset Fr0, Fe0.

**Figure 8 sensors-23-06654-f008:**
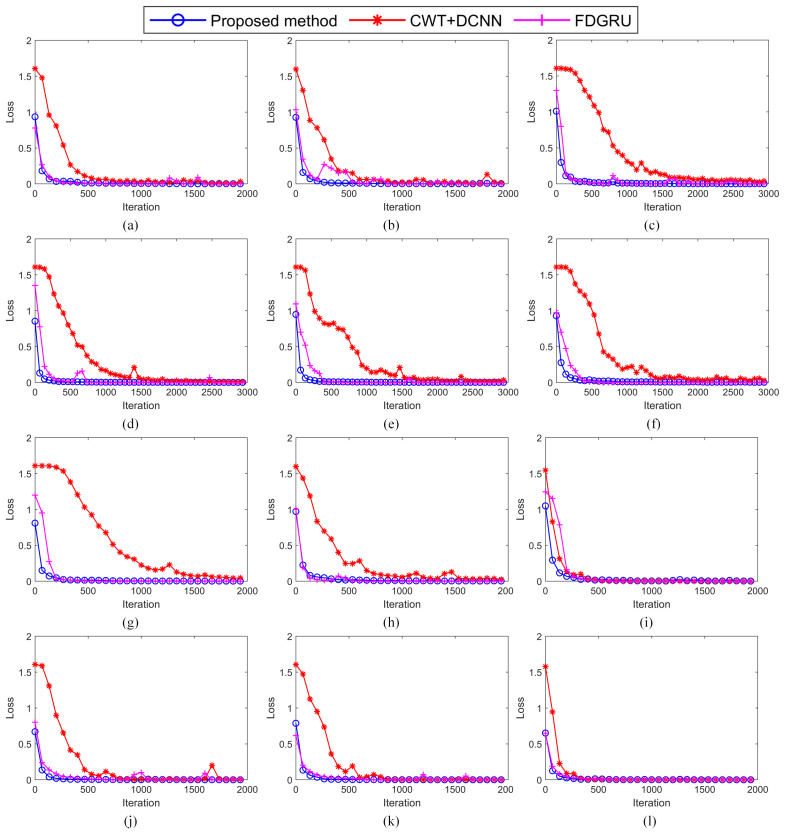
Curves of training error of three methods. (**a**) Dataset Sr0. (**b**) Dataset Sr1. (**c**) Dataset Sr2. (**d**) Dataset Sr3. (**e**) Dataset Sr4. (**f**) Dataset Sr5. (**g**) Dataset Sr6. (**h**) Dataset Sr7. (**i**) Dataset Mr0. (**j**) Dataset Mr1. (**k**) Dataset Mr2. (**l**) Dataset Fr0.

**Figure 9 sensors-23-06654-f009:**
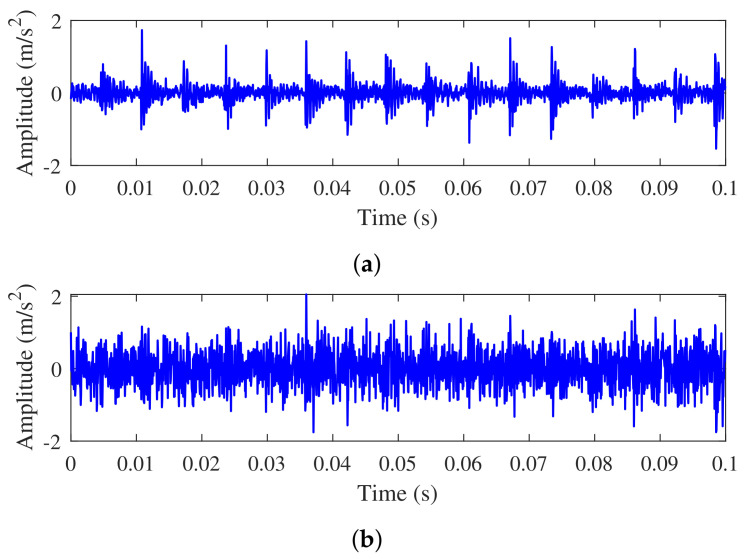
Time domain waveforms for original signal of inner race fault and the composite noisy signal with SNR = 0 dB, respectively. (**a**) Original signal. (**b**) Noisy signal.

**Figure 10 sensors-23-06654-f010:**
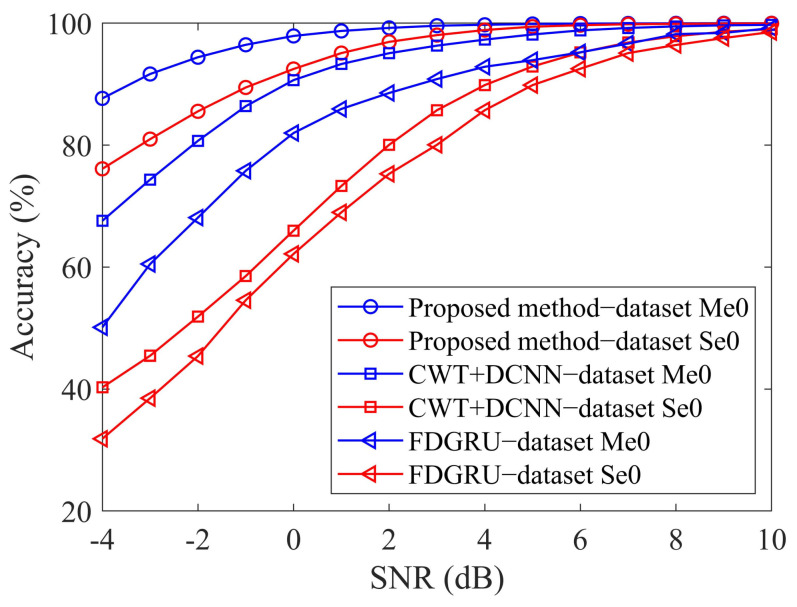
Fault diagnosis results of different methods varying with SNRs.

**Figure 11 sensors-23-06654-f011:**
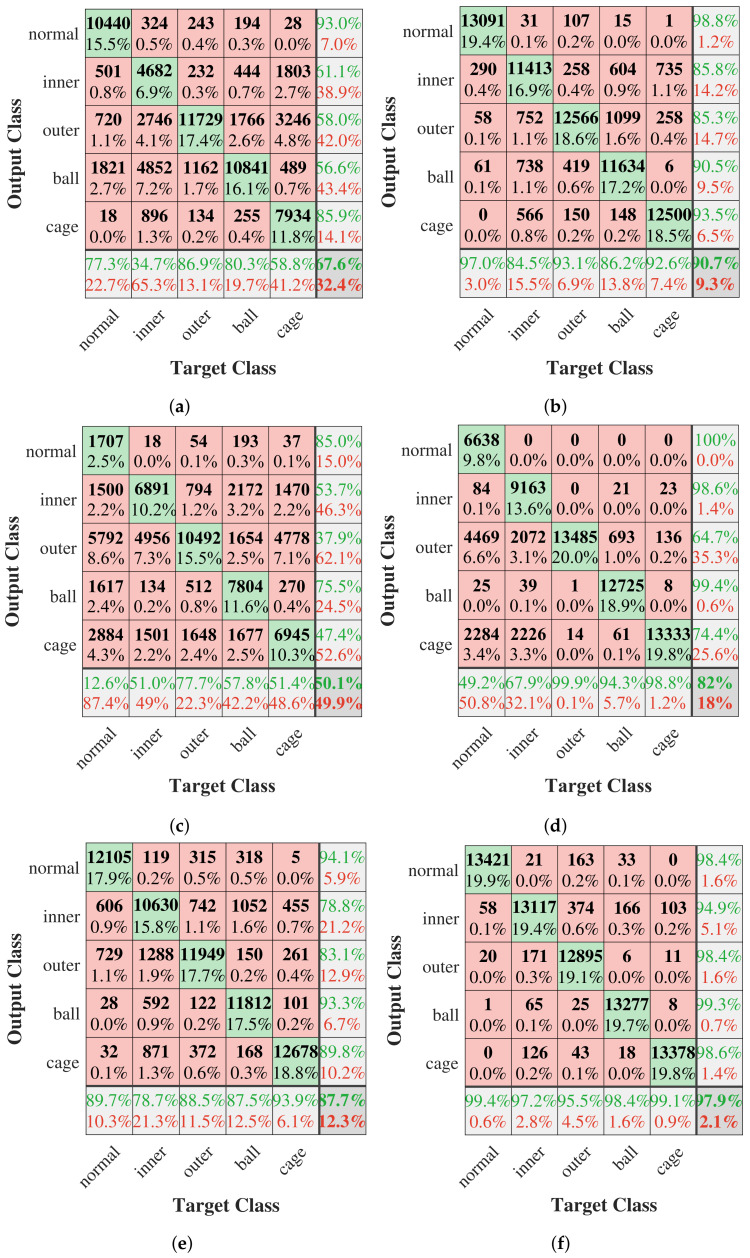
Confusion matrix of the three methods in dataset Me0. (**a**) SNR = −4 dB, CWT+DCNN algorithm. (**b**) SNR = 0 dB, CWT+DCNN algorithm. (**c**) SNR = −4 dB, FDGRU method. (**d**) SNR = 0 dB, FDGRU method. (**e**) SNR = −4 dB, the proposed method. (**f**) SNR = 0 dB, the proposed method.

**Figure 12 sensors-23-06654-f012:**
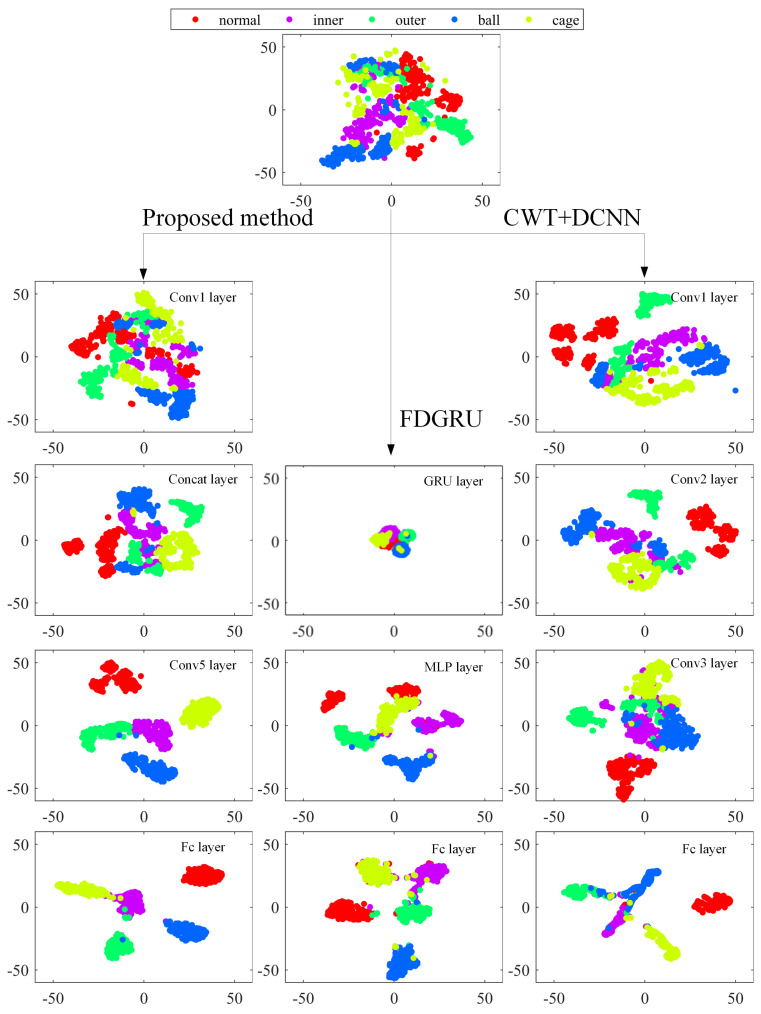
Feature visualization via t-SNE: feature representations for partial samples of dataset Me0 and their feature expression in MCFNN, DCNN and FDGRU.

**Table 1 sensors-23-06654-t001:** Details of the proposed MCFNN model used in experiments.

No.	Layer Type	Kernel Size	Stride	Kernel Number	Output Size	Padding
1	Conv1	3 × 3 × c	2 × 2	32	31 × 31 × 32	No
2	Conv2	1× 1 × 32	2 × 2	64	16 × 16 × 64	No
3	Conv3 (with dropout)	3 × 3 × 32	2 × 2	64	16 × 16 × 64	Yes
4	Conv4 (with dropout)	5 × 5 × 32	2 × 2	64	16 × 16 × 64	Yes
5	Add1	-	-	-	16 × 16 × 64	-
6	Add2	-	-	-	16 × 16 × 64	-
7	Concat	-	-	-	16 × 16 × 128	-
8	Conv5	3 × 3 × 128	1 × 1	96	14 × 14 × 96	No
9	Pool1	2 × 2	1 × 1	-	7 × 7 × 96	No
10	Conv6	3 × 3 × 96	2 × 2	32	3 × 3 × 32	No
11	Pool2	2 × 2	1 × 1	-	1 × 1 × 96	No
12	Softmax	5	1	1	1 × 5	No

**Table 2 sensors-23-06654-t002:** The size parameters of bearing.

Internal Diameter (mm)	External Diameter (mm)	Width (mm)	Number of Balls	Ball Diameter (mm)	Pitch Diameter (mm)	Preset Contact Angle (°)	Actual Contact Angle (°)
142.9	190.0	33	17	24.6	166.45	25.4∼31.5	28∼56

**Table 3 sensors-23-06654-t003:** Description of bearing datasets.

	Name of Dataset		Loads (kN)	Including Channels		Speed (rpm)			Sample Size
	Single Channel	Multi-Channel		Single Channel	Multi-Channel	Range (rpm)	Step (rpm)	Number of Rotational Speeds	
Training set	Sr0/Sr1/ Sr2/Sr3 Sr4/Sr5/ Sr6/Sr7	Mr0/Mr1/ Mr2/Fr0	4~9	AC0/AC1/ AC2/AC3 AC0/AC1/ AC2/AC3	AC0, AC1, AC7/ AC2, AC3, AC4/ AC5, AC6, AC7/ All channels	1000~10,000	200	46	Sr: 64 × 64 × 1 Mr: 64 × 64 × 3 Fr: 64 × 64 × 8
Test set	Se0/Se1/ Se2/Se3 Se4/Se5/ Se6/Se7	Me0/Me1/ Me2/Fe0	4~9	AC0/AC1/ AC2/AC3 AC0/AC1/ AC2/AC3	AC0, AC1, AC7/ AC2, AC3, AC4/ AC5, AC6, AC7/ All channels	1100~9900	200	45	Se: 64 × 64 × 1 Me: 64 × 64 × 3 Fe: 64 × 64 × 8

**Table 4 sensors-23-06654-t004:** The input samples generated under various parameters.

No.	Window Length	Overlapping Ratio (%)	FFT Points	Time-Frequency Resolution	Input Sample Size
1	64	3.13	32	17 × 17	16 × 16 × 3
2	128	7.81	32	17 × 17	16 × 16 × 3
3	64	6.25	64	33 × 33	32 × 32 × 3
4	128	53.13	64	33 × 33	32 × 32 × 3
5	64	53.13	128	65 × 65	64 × 64 × 3
6	128	77.34	128	65 × 65	64 × 64 × 3
7	128	88.28	256	129 × 129	128 × 128 × 3

**Table 5 sensors-23-06654-t005:** The influence of learning rate on training and test results by MCFNN.

Learning Rate	Training Average Loss Function	Test Average Function	Training Average Accuracy (%)	Test Average Accuracy (%)	Average Training Time (s)
0.0001	0.0006	0.0018	100	99.98	76
0.0005	0.0010	0.0016	100	100	71
0.001	0.0007	0.0012	100	100	43
0.005	0.0010	0.0028	100	99.93	48
0.010	0.0005	0.0044	100	99.84	51
0.015	0.0003	0.0049	100	99.84	54
0.020	0.0009	0.0120	100	99.78	53
0.030	0.0010	0.0242	100	99.53	56

**Table 6 sensors-23-06654-t006:** Comparison of accuracy (mean accuracy ± standard deviation).

Dataset	Proposed Method		CWT+DCNN Algorithm		FDGRU	
	Training Accuracy (%)	Testing Accuracy (%)	Training Accuracy (%)	Testing Accuracy (%)	Training Accuracy (%)	Testing Accuracy (%)
(Sr0, Se0)	98.40 ± 1.66	98.25 ± 1.76	97.82 ± 0.71	97.93 ± 0.80	97.75 ± 1.14	97.63 ± 1.22
(Sr1, Se1)	98.46 ± 0.97	98.26 ± 1.00	97.83 ± 0.90	97.69 ± 0.91	97.65 ± 0.86	97.57 ± 0.86
(Sr2, Se2)	95.44 ± 2.42	93.35 ± 2.19	68.91 ± 4.62	67.82 ± 4.61	71.13 ± 5.74	70.86 ± 5.78
(Sr3, Se3)	98.90 ± 0.32	96.68 ± 0.42	77.03 ± 3.86	76.88 ± 3.29	76.96 ± 3.50	76.62 ± 3.58
(Sr4, Se4)	97.54 ± 0.73	96.42 ± 0.78	75.12 ± 5.32	73.05 ± 5.17	74.84 ± 5.19	74.76 ± 4.88
(Sr5, Se5)	97.74 ± 1.07	96.80 ± 0.75	81.80 ± 4.14	80.84 ± 4.45	80.34 ± 3.18	79.57 ± 3.47
(Sr6, Se6)	99.16 ± 0.71	97.63 ± 0.65	95.77 ± 2.77	94.26 ± 2.89	95.46 ± 2.03	94.18 ± 1.73
(Sr7, Se7)	98.77 ± 0.97	98.41 ± 0.96	95.40 ± 1.61	95.31 ± 1.60	95.11 ± 1.59	94.75 ± 1.52
(Mr0, Me0)	100	100	99.60 ± 0.86	99.28 ± 1.71	99.50 ± 0.52	99.32 ± 0.54
(Mr1, Me1)	100	99.91 ± 0.09	99.98 ± 0.04	99.41 ± 0.20	99.93 ± 0.12	99.50 ± 0.26
(Mr2, Me2)	100	99.90 ± 0.08	99.92 ± 0.17	99.74 ± 0.21	99.91 ± 0.13	99.55 ± 0.24
(Fr0, Fe0)	100	100	99.98 ± 0.05	99.97 ± 0.06	99.96 ± 0.09	99.93 ± 0.14

## Data Availability

Not applicable.

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
