# Peer review of "An Anti-Noise Convolutional Neural Network for Bearing Fault Diagnosis Based on Multi-Channel Data"

_sensors, 2023, doi:10.3390/s23156654_

Round 1
Reviewer 1 Report
In this paper, multi-channel technology and convolutional neural network fusion are used to improve the anti-noise ability of rolling bearing fault diagnosis, and dropout algorithm is used to improve the robustness of the model. The structure of the article is more rigorous, the method is feasible, and the experimental verification results are better, but there are still the following points that need to be supplemented or explained by the author.
1 ) At present, fault diagnosis using convolutional neural network technology with vibration signals as feature samples mostly adopts two general methods : transforming vibration signals into time-frequency images and directly convolutional classification of signal data. Does the author need to discuss whether the latter has better effect ?
2 ) The vibration signal sample size of the article is selected as a sample of 0.1 seconds. Does the author need to explain the rationality of this setting ? Under different working conditions, speed and sampling frequency of rolling bearings, is the 0.1 second data sample size suitable ?
3 ) The author compares the improvement of fault diagnosis accuracy after the fusion of multi-channel technology and convolutional neural network compared with single-channel convolutional neural network through experiments. However, to prove the superiority of the proposed model, please supplement the comparison results with other published excellent models.
4 ) It can be seen from the T-SNE diagram that the method proposed in this paper is not very good in the distinction between the cage fault and the inner ring fault of the rolling bearing. On the contrary, the CWT-DCNN model compared in this paper is excellent in the distinction between the cage fault and the inner ring fault of the rolling bearing. How to explain this phenomenon ?
5 ) The most common problem of rolling bearing fault diagnosis is the long tail distribution problem of fault data scarcity. The results of this paper are obtained under balanced samples. Can we supplement and discuss the fault diagnosis effect of the model under unbalanced samples such as more normal data and less fault data, or discuss whether oversampling, undersampling and other technologies can improve the performance of the model ?
Minor editing of English language required
Reviewer 2 Report
The authors proposed an anti-noise convolutional neural network for bearing fault diagnosis based on multi-channel data to improve the anti-noise performance of neural network. The motivation of this research is very interesting, while the organization of this paper needs to be improved greatly. The comments and suggestion of this reviewer lie the following aspects:
-1-. Explain the multi-channel data selection construction criteria.
-2-. Since the main purpose of this paper is to propose an anti-noise convolutional neural network for fault diagnosis, please explain the purpose of experimental verification of some noise-free environments.
-3-. In Figure 12, the final clustering effect of the proposed method is partially coincident, and can it be further optimized.
The presentation can be improved.
Reviewer 3 Report
1- Please enhance the abstract by adding some prominent results.
2- The authors must enhance the Motivation section and discuss more related issues to it.
3- More description should be added for the methodology
4- The Introduction section must be updated with recently published papers such as: Improved double-surface sliding mode observer for flux and speed estimation of induction motors;
5- The contribution section should be written more clearly and highlight the strengths of the paper.
6- The conclusion section can be updated with prominent and numerical results.
Round 2
Reviewer 3 Report
accepted